# Potential Protective Role of Pregnancy and Breastfeeding in Delaying Onset Symptoms Related to Multiple Sclerosis

**DOI:** 10.3390/medicina59030619

**Published:** 2023-03-20

**Authors:** Alessandra Logoteta, Maria Grazia Piccioni, Riccardo Nistri, Laura De Giglio, Valentina Bruno, Giuseppe La Torre, Stefano Ianni, Luana Fabrizi, Ludovico Muzii, Carlo Pozzilli, Serena Ruggieri

**Affiliations:** 1Department of Maternal Infantile and Urological Sciences, Sapienza University of Rome, 00161 Rome, Italy; 2Department of Human Neurosciences, Sapienza University of Rome, 00185 Rome, Italy; 3Neurology Unit, Medicine Department, San Filippo Neri Hospital, 00135 Rome, Italy; 4Gynecologic Oncology Unit, Department of Experimental Clinical Oncology, IRCCS “Regina Elena” National Cancer Institute, 00144 Rome, Italy; 5Obstetrics and Gynecology, Saint Camillus International University of Health and Medical Sciences, 00131 Rome, Italy; 6Department of Public Health and Infectious Diseases, Sapienza University of Rome, 00185 Rome, Italy; 7Department of Anesthesiology, Critical Care Medicine and Pain Therapy, Sapienza University of Rome, 00185 Rome, Italy; 8Department of Experimental Clinical Oncology, Anesthesia, Resuscitation, and Intensive Care Unit IRCCS “Regina Elena” National Cancer Institute, 00144 Rome, Italy

**Keywords:** multiparity, late-onset multiple sclerosis, pregnancy, breastfeeding, hormone replacement therapy, disability, neuroimmunology

## Abstract

The impact of pregnancy and breastfeeding on the development and outcomes of Multiple sclerosis (MS) has been debated for decades. Since several factors can influence the evolution of the disease, the protective role of multiparity and breastfeeding remains uncertain, as well the role of hormone replacement therapy in the perimenopausal period. We report two cases of relatively late-onset MS in two parous women, who developed their first neurological symptoms after six and nine pregnancies, respectively. Both women breastfed each of their children for 3 to 12 months. One of them underwent surgical menopause and received hormone replacement therapy for 7 years before MS onset. We performed a systematic literature review to highlight the characteristics shared by women who develop the disease in similar conditions, after unique hormonal imbalances, and to collect promising evidence on this controversial issue. Several studies suggest that the beneficial effects of pregnancy and breastfeeding on MS onset and disability accumulation may only be realized when several pregnancies occur. However, these data on pregnancy and breastfeeding and their long-term benefits on MS outcomes suffer from the possibility of reverse causality, as women with milder impairment might choose to become pregnant more readily than those with a higher level of disability. Thus, the hypothesis that multiparity might have a protective role on MS outcomes needs to be tested in larger prospective cohort studies of neo-diagnosed women, evaluating both clinical and radiological features at presentation.

## 1. Introduction

Multiple sclerosis is a chronic demyelinating autoimmune neurodegenerative disease of the central nervous system, usually occurring in young adults with a female/male prevalence of approximately 3:1 [1].

The disease course can either be relapsing-remitting, characterized by acute inflammatory episodes of CNS damage (relapse) and subsequent remissions, or progressive. Relapsing-remitting MS is characterized by an early focal inflammation phase which leads to neurodegenerative insult during each relapse, with substantial clinical stability in intervals between relapses, whereas in progressive multiple sclerosis, the predominant mechanisms of neurodegeneration are observed from the onset of the disease and progressive disability worsening occurs despite no clinically identifiable episodes of CNS demyelinating injury. Patients with MS acquire disability either through relapse-associated worsening or a deterioration process that does not depend on relapse activity in the progressive form [2,3]. In Italy, at least 70–75% of patients are affected by the relapsing-remitting form of MS (RR-MS) [4]. Disease activity is determined clinically by the frequency and severity of relapses and/or with the use of magnetic resonance imagining (MRI) by means of the detection of new or enlarged brain and spinal cord lesions, while disability worsening is assessed according to a clinical score, the Expanded Disability Status Scale [5]. 

Women who are diagnosed with MS at reproductive age and express a desire for parenthood should be carefully counselled for pregnancy planning, and each patient’s MS history and current neurologic deficits should be considered independently [6].

The knowledge regarding multiple sclerosis and the pregnancy paradigm has evolved over time. In the past, women with MS were advised to avoid pregnancy altogether based on data from case reports and small case studies [7], but more recently, a pivotal study regarding pregnancy in MS documented a significant reduction in the relapse rate during pregnancy, followed by a sharp increase in the number of relapses in the immediate postpartum period [8]. 

At present, it is known that multiple sclerosis, as well as many other autoimmune diseases, benefits from the immune activity suppression induced by pregnancy [9], but whether childbirth has a beneficial effect on the long-term disease course in women with MS is still a matter of debate. Pregnancy’s impact in MS has been reported to be beneficial, delaying disability progression [10,11,12], or at least neutral, when adjusting for confounding factors [13,14]. The progressive introduction of preventive therapies (or disease modifying drugs (DMDs)) complicated this scenario, due to the fact that most of them are potentially the cause of harmful effects on the embryo or fetus. Hence the necessity for pregnancy planning, regarding both the course of the disease and the therapy: through the pre-conception, pregnancy and post-partum periods, there is a need for disease control management to decrease chances of MS relapses while avoiding the potential risks to the mother and the fetus. Disease-modifying treatment during pregnancy needs to be adjusted individually, taking into consideration the patient’s priorities, age, severity of disability, clinical and MRI disease activity, relapse rate and the risk/benefit ratio of continuing the treatment. According to the US Food and Drug Administration (FDA), most of the drugs registered to treat MS are labeled as class C, meaning that data relating to adverse effects were obtained in animal reproduction studies; however, despite the possible risk, pregnant women may benefit from these drugs, but there is not enough reliable evidence and a lack of well-controlled studies in humans. It is worth noting that the only safe DMD in pregnancy is glatiramer acetate 20 mg/mL, labeled as “B” [15]. It appears that this counselling could represent a real challenge even for expert clinicians, as it may be difficult to find the right balance between the risk of relapse and the need for DMDs which impact both maternal health and pregnancy outcomes [6]. For the purpose of this review, we specify that dimethyl fumarate and fingolimod are both class C drugs, but dimethyl fumarate has a short half-life and does not require a washout period, and in human studies the results of pregnancy outcomes when exposed to dimethyl fumarate early in pregnancy did not indicate increased fetal abnormalities. On the other hand, fingolimod should not be administered due to its effect on the receptors responsible for vascular system formation during the sensitive period of embryogenesis. Two months are required for fingolimod to be eliminated from the body, so during this period, contraception should be maintained. The fingolimod clinical program analyzed pregnancy outcomes after exposure to the drug at the time of conception or 6 weeks before: 20 elective terminations, 4 terminations due to fetal abnormalities, 9 spontaneous abortions, and 2 fetuses born with malformations were reported [16]. 

We present two cases of women who developed the first symptoms of MS at 51 and 43 years old, respectively, suggesting that their multiparity could have delayed the onset of their symptoms. These clinical cases are reported as per the CARE checklist guidelines [17]. We performed a systematic literature review considering the diagnosis and pathophysiology of the multiparity condition; the search was extended up to October 2022 by the authors independently, covering several databases (MEDLINE, PubMed, Embase). 

## 2. Case Presentation

### 2.1. Case Report 1

The first case study was a 51-year-old woman of perimenopausal age, without any relevant medical history except for allergic asthma in her childhood and allergies to acetylsalicylic acid, cow’s milk, mites and dust. At a younger age, she underwent cataract surgery and cholecystectomy. The patient reported no neurological symptoms throughout her life. 

Her first menstrual period occurred at the age of 14 and she got pregnant 7 times, from 1993 to 2004, giving birth to 6 females (1 spontaneous abortion). All 6 daughters were breastfed for an average period of six months each. She underwent a hysterectomy in 2004 for uterine fibroids and in 2009 she underwent a bilateral annessectomy for ovarian cysts with the consequent onset of iatrogenic menopause. From 2009 to 2016, she was treated with hormone replacement therapy (HRT) with tibolone, a selective tissue estrogenic activity regulator. 

In March 2017, she presented with motor impairment of her left arm with difficulties in fine hand movements. Brain and spinal cord magnetic resonance imaging (MRI) was performed five days after the onset of clinical symptoms and showed multiple hyperintense lesions in the periventricular white matter on the T2-weighted sequences, particularly in the peri-frontal area, involving the head of the caudate nucleus; all lesions showed contrast enhancement after gadolinium administration (Figure 1). Spinal cord imaging also revealed several T2 hyperintense contrast enhancing areas in the posterolateral left tract from C2 to D1 (Figure 1). The patient was treated with 1 g of methylprednisolone i.v. for 5 days with partial regression of symptoms and then was admitted to the day hospital in order to perform a lumbar puncture. A cerebrospinal fluid (CSF) exam showed 5 oligoclonal bands. A polymerase chain reaction search on the CSF for neurotropic viruses was negative. Neurological examination showed a positive Romberg test, and a mild strength deficit in both legs. Hypoesthesia of the left arm and hand was observed, with the medullary sensory level at D8 and urinary urgency, with an Expanded Disability Status Scale (EDSS) score of 2.5. After performing all the immunological and infectious diagnostic tests necessary to rule out other possible diagnoses, the patient was therefore diagnosed with RR-MS according to the McDonald Criteria 2010 in June 2017 [18]. The diagnosis was based on the evidence of a clinical episode such as the one described above and the satisfaction of dissemination in space (two lesions in at least two of the four locations considered characteristic of MS—juxtacortical, periventricular, infratentorial, and spinal cord) and in time criteria, as there were both enhancing (not only symptomatic) and non-enhancing lesions in the MRI. 

The patient started preventive treatment with dimethyl fumarate in June 2017. During clinical and radiological follow-up, the patient did not experience either clinical relapse or new T2 or gadolinium-enhancing lesions. The last neurological examination in March 2022 was stable compared to the previous assessment and the patient did not complain of new or worsening symptoms. Preventive treatment with dimethyl fumarate was still ongoing at the last neurological visit. 

### 2.2. Case Report 2

The second case study was a 41-year-old patient of childbearing age; she had no relevant clinical history, except for a nickel allergy and a smoking habit, at about 10 cigarettes per day in the last 20 years. The patient reported no neurological symptoms throughout her life. 

Born by vaginal delivery, her menarche occurred at the age of 12, and she had nine full-term pregnancies and two abortions (one spontaneous and one voluntary) from the age of 16 to 39 years old. All of her six daughters and three sons were breastfed for a period ranging from 3 months to 1 year. 

In November 2018, she presented with weakness in both legs. Due to the persistence of this symptom, the patient was admitted to the hospital, where she underwent several exams. Brain and spinal cord MRI showed on T2-weighted sequences multiple hyperintense lesions localized in the periventricular white matter, corona radiata bilaterally, brainstem, right cerebellar superior peduncle; one of these lesions on the parietal lobe showed Gd-enhancement (Figure 2). Spinal cord imaging also revealed one T2 hyperintense lesion area in the posterior tract from C1 to C3 (Figure 2). 

The patient was treated with 1 g of methylprednisolone i.v. for 5 days with partial regression of symptoms. The CSF exam showed several oligoclonal bands. Neurological examination showed a moderate motor deficit in both legs, modest difficulties in alternate movements in both arms, a positive Romberg test with slight ataxia and urinary urgency. The EDSS was 3.5. The patient underwent all diagnostic tests to assure no better explanation. No immunological or infectious abnormalities were detected. Therefore, a diagnosis of RR-MS was provided in December 2018 according to modified McDonald Criteria 2017 [19]. Indeed, the patient presented with a clinical episode and the presence of dissemination in space (multiple T2 lesions detected in several CNS regions) as well dissemination in time criteria (enhancing and no enhancing lesions and the presence of an oligoclonal band). 

The patient started preventive treatment with fingolimod in January 2019. During clinical and radiological follow-up, the patient did not experience either clinical relapse or new T2 or gadolinium-enhancing lesions. The last neurological examination, in March 2020, was stable compared to the previous assessment and patient did not complain of new or worsening symptoms. Treatment with fingolimod was ongoing at last contact. 

## 3. Literature Review and Discussion 

The two cases described, although anectodical, highlight the putative protective role of multiparity and, potentially, of breastfeeding and HRT in MS. The first patient, at 51 years old, with six daughters, developed a highly active form of MS after the interruption of HRT that she had been receiving for 7 years. The second patient, at 43 years old, received a diagnosis of MS soon after her ninth pregnancy. We can speculate that both women possibly developed clinical MS due to the decrease in the level of estrogen after it had been maintained at a very high level for a long period of time. Starting from these two cases, we can broaden our considerations, pointing out the possible role of parity in the course of MS, summarizing the studies and the considerations that have addressed this topic during recent years in the scientific literature (Table 1). 

### 3.1. Pregnancy 

The most widely accepted theory to explain the protective effect of estrogen and other sex hormones is that they may induce immunological changes by shifting the T helper (Th) cell profile to predominantly Th2 (anti-inflammatory cytokines) rather than Th1 and Th-17 (pro-inflammatory cytokines) [32,33]. Maternal estrogens and other sex hormones are responsible for the immunological shift and increased levels of circulating regulatory T cells (Treg) cells. It is also known that estrogens have an anti-inflammatory and neuroprotective effect themselves [7,34,35]. In addition, the development of maternal–fetal immunotolerance in pregnancy seems to be associated also with a shift from cell-mediated immunity toward humoral immunity. The fetal–placental unit secretes cytokines such as interleukin-10 that down-regulates maternal cellular immunity response [34,35,36].

Sex hormones’ fluctuations during pregnancy are unparalleled by any other neuroendocrine events (e.g., menstruation, puberty, menopause); such a massive hormonal production can be largely attributed to placental secretion of estrogens and progesterone, which rise to 20-fold during pregnancy and plummet after delivery [36].

This physiology could explain why, in T-cell–mediated autoimmune diseases such as MS, pregnancy appears to be associated with the spontaneous remission of disease activity, while the post-partum period is associated with clinical exacerbations. 

In animal models of MS, such as experimental allergic encephalomyelitis (EAE), significant symptomatic improvements are seen during pregnancy across a range of species [37,38]. For instance, in mice, a combination of estrogen, placental lactogen and other hormones of late pregnancy reduce EAE activity, also showing a beneficial effect on the endogenous repair of CNS demyelinated lesions, on the differentiation of new oligodendrocytes and on remyelination [37,39] (see Figure 3).

In 2020, Deems and Leuner [40] reviewed the effect of the post-partum period and parity on a series of CNS pathologic conditions. They emphasized the influence of estrogens (E3 in particular, predominant in late pregnancy), via the estrogen receptor alpha (ERα) and partially beta (ERβ), progesterone and, in a smaller measure, prolactin, on increased functional remyelination and the protection of existing axons and a significant reduction in CNS inflammation. The long-term effect of parity on disability progression requires further research, but eventually, all the beneficial effects of hormonal changes should result in a different disease course. 

As a side note, parity accounts for lasting changes in immune function and CNS, and one way these could arise is through epigenetic effects. A gene-wide association study of differentially expressed genes in parous versus non-parous women with MS identified 574 genes that changed with parity and overlapped with MS genes in humans and rat models. Around 40% of these overlapping genes were pertinent to axonal guidance or developmental biology and cell-to-cell communication pathways and were significantly upregulated in parous women as compared to nulliparous women with MS [41], suggesting that parity may alter epigenetic mechanisms to enhance the repair of damaged axons. 

As previously mentioned, the first study ever to support this evidence was the multicentric observational Pregnancy in Multiple Sclerosis (PRIMS) study, which involved 12 European countries and prospectively analyzed 269 pregnancies in 254 women with RR-MS 12 months after delivery, comparing the course of the disease over the year prior to conception and the year following childbirth. The primary endpoints were the determination of the annualized relapse rate (ARR) and the progression of disability [8]. In this study, Confavreux and colleagues provided evidence that the relapse rate decreases during the first and the second trimester of pregnancy and in the first three months post-partum as compared to the year before pregnancy. ARR ranged from 0.7 in the year before pregnancy (*p* = 0.03 in comparison with the rate before pregnancy) to 0.5, 0.6 and 0.2, respectively, in the first, second and third trimesters of pregnancy (*p* < 0.001). The ARR increased to 1.2 during the first three months post-partum (*p* < 0.001), but then returned to the pre-pregnancy rate. Women who breastfed their infants had a significantly lower rate of relapse than women who did not. Authors also collected data on pregnancy and delivery outcomes such as delivery mode (vaginal vs. cesarean), timing (preterm vs. term), complications and birthweight, and concluded that none of these was influenced by MS. 

This paper, even though it was published before the era of DMDs, set a milestone for studies ahead, as for the first time the protective role of pregnancy was described, showing a reduction in relapse rate even more marked than any therapeutic effect reported at that time. Two key points emerged from this study which are still important to keep in mind during patient counselling: the global ARR referring to the year of pregnancy (9 months of gestation + 3 months after childbirth) is comparable to pre-pregnancy ARR, and only about one-third of women (28%) have a post-partum relapse.

Given the evidence of a rebound of disease activity, Vukusic and colleagues conducted a follow-up survey of the PRIMS study. They collected data based on a 2-year post-partum follow-up of 227 women to define clinical factors which could predict the probability of a relapse in the 3 months after the delivery [21]. Again, the authors concluded that compared to the pre-pregnancy year, the relapse rate decreased significantly during pregnancy, especially in the third trimester, and increased in the first trimester after delivery. From the second trimester after delivery onwards, in the 2 years post-partum, the relapse rate did not differ from that in the pre-pregnancy year (ARR ranging from 0.6 to 0.5 (*p* 0.4 95% CI 0.4–0.6 CI) vs. 0.7 in the year pre-pregnancy). Multivariate analysis showed that there were only two factors that correlated with the likelihood of relapse and new disease activity in the first 3 months after delivery: disability progression (EDSS) and the number of relapses, respectively, during pregnancy and in the year before [20,22]. The ARR of the pregnancy year, defined as encompassing 12 months, the nine months of pregnancy and the first 3-month period after delivery, was similar to that of the pre-pregnancy year.

In 2011, Finkelsztejn and colleagues published a meta-analysis covering 22 original studies with a total of 13,144 pregnancies [23]. The ARR during the year preceding pregnancy was 0.4 relapses/year (95% CI 0.40–0.72), 0.2 (95% CI 0.19–0.32) during pregnancy, and after the delivery, it peaked at 0.7 relapses/year (95% CI 0.64–0.87). They also collected data on breastfeeding, pregnancy complications, delivery mode, risk of prematurity and malformation, and concluded that women with MS do not present a significantly higher risk of obstetrical complications. 

In an Italian study by Portaccio and colleagues, relapses and recurrence appeared to be predicted only by disease activity before and during pregnancy, while a lower risk of postpartum relapse was associated with the use of DMDs in the two years prior to conception [24]. A few years later, a large study on 893 pregnancies in 674 females with RRMS receiving DMDs demonstrated that ARR pre-pregnancy, calculated to be 0.32, fell to 0.13 in the third trimester and rose to 0.61 in the first three months post-partum. Median EDSS did not vary, confirming that exposure to DMDs before conception significantly reduces the risk of post-partum relapse [26]. Other studies in the DMD era have also confirmed that the relapse rate decreases dramatically during pregnancy and increases slightly in the early post-partum period [42]. 

In a large systematic review on 2466 patients worldwide [11], a total of 304 full-term pregnancies were reported for 226 (12.3%) females over a 10-year observation period, of which 134 (44.1%) were conceived whilst receiving therapy. Pregnancies were independently associated with a median 3.17-point lower and a median 0.36-point decrease in EDSS score over the 10 years of follow-up, and any time spent pregnant (including induced and spontaneous abortions) was beneficial. The authors indeed reported that the therapeutic effect of pregnancy was more than four times greater than that of first-line therapy in women within the first 10 years of starting DMDs. 

Further research has confirmed that the MS relapse rate increases up to 6 months postpartum and, when considering DMD use, although uncommon in the year before pregnancy, it decreased immediately pre-pregnancy and during pregnancy [27]. 

Dobson and Giovannoni [28] published a metanalysis of updated evidence since the PRISM study including 28 studies and a total of 7034 MS pregnancies in 6430 women. ARR fell from 0.57 pre-pregnancy to, respectively, 0.36, 0.29, and 0.16 during the first, second and third trimesters with a post-partum rebound (ARR 0.85), thus confirming the historic assumption that ARR reduces during pregnancy. On the other hand, this analysis did not show a relationship between the pre-pregnancy relapse rate and DMD exposure, but evidenced for the first time an attenuation in the post-partum rebound of relapse rate that may be due to the early resumption of DMDs.

These clinical observations are corroborated by a documented cessation of disease activity on MRI during the third trimester of pregnancy, in terms of gadolinium enhancement and new lesions [43].

A recent meta-analysis [30] on 23 prospective/retrospective cohort studies measuring changes in the ARR during pregnancy and puerperium found a significant pooled mean reduction during pregnancy compared to the pre-pregnancy year in the ARR for 15 cohorts, and a mean increase in the ARR in the 3-month puerperium relative to the pre-pregnancy year for 14 cohorts included in the meta-analysis. Disability worsening was addressed in 18 cohorts, and in 14 of them, there were no significant changes. Peripartum complications and obstetrical outcomes were assessed in 16 cohorts without finding significant differences. 

The latest systematic review on relapse rates during pregnancy and post-partum in patients with relapsing MS addressed the question of DMDs administration, preconceptionally and in the post-partum period. The review included 28 studies and 4739 pregnancies in 5324 patients. All five studies comparing high-efficacy DMDs use preconception or no DMDs suggested that there was a greater risk of relapse during pregnancy following the withdrawal of high-efficacy DMDs. Of the 10 studies evaluating relapses during pregnancy, five studies found that continuing DMDs into early pregnancy reduced relapses compared to discontinuing treatment. DMDs exposure during pregnancy was associated with fewer postpartum relapses versus no DMDs exposure in four out of seven studies, while three found no difference between groups. The authors concluded that DMDs administration for effective disease-management strategies in these especially high-risk patients must be carefully evaluated [31].

It has been conveniently hypothesized that stress, immune system imbalances, together with hormone withdrawal following delivery and loss of the immunosuppressive state acquired in pregnancy, may indeed account for the rebound of clinical and radiological disease activity during the first three months of the post-partum period [44]. 

More controversial is whether pregnancy can delay the first episode of demyelination or a confirmed diagnosis of MS. 

Although some studies showed that parity did not impact MS, there is also evidence that higher parity and offspring number are associated with a reduced risk of a first demyelinating event, suggesting a potential suppressive effect [10,45,46,47,48].

This latter hypothesis was recently reinforced by an extensive retrospective study based on the MSBase registry, in which authors collected data on reproductive history (duration of each pregnancy, date of delivery, length of breastfeeding) from 2557 women with MS. The authors assessed that women with previous pregnancies and childbirths had a later onset of clinically isolated syndrome (CIS) compared with those who had never been pregnant (HR, 0.68; 95% CI, 0.62–0.75; *p* < 0.001), with a median delay of 3.3 (95% CI, 2.5–4.1) years. Women who had given birth also had a later CIS onset compared with women who had never given birth (Hazard Ratio (HR) 0.68; 95% confidence interval (CI), 0.61–0.75; *p* < 0.001), with a similar median delay of 3.4 (95% CI, 1.6–5.2). However, higher gravidity and parity number were not associated with a delay in CIS onset [49]. 

### 3.2. Disability Progression

Previous studies supported the beneficial effect of pregnancy on long-term disability, in particular the association between past pregnancy, offspring number, and risk of a first episode of demyelination, demonstrating that higher parity was associated with a reduced risk of such events [47,48,50]. However, the reverse causality (i.e., reduced reproductive activity in persons with yet-undiagnosed MS) has been demonstrated to explain the observed associations [51]. Later, a large cohort study based on data from 1195 parous women compared with 328 nulliparous women with MS concluded that parous women took a longer time to reach EDSS 6 than nulliparous women [25]. 

Yet another retrospective study compared the time taken to reach an EDSS score of 4 and an EDSS of 6 between parous women and nulliparous women with MS. The study demonstrated that after clinical onset, multiparous women had a lower risk and took a longer time to reach typical disability milestones (EDSS of 4 and 6) [12] than monoparous women, including the effect of post-partum relapses, despite the evidence that nulliparous women had a shorter mean disease duration and were more often treated with a second-line drug than parous women [12]. Other works have reported that the beneficial effects of pregnancy on delaying disability accumulation may only be realized with two or more pregnancies [10]. The abovementioned large study from MSBase using 10-year follow-up data on 2466 women demonstrated the protective effect of pregnancy to be more than four times more potent than first-line DMDs (interferon-beta and glatiramer acetate) in preventing long-term disability accrual [11]. Interestingly, some authors found that comparing the proportion of time spent pregnant to the proportion of time on first-line therapy, the therapeutic effect of pregnancy is more effective than that obtained under therapy, thus demonstrating that pregnancy could be beneficial in women with MS, especially in those with a lower disability burden and relapse rate [11]. 

On the other hand, a Canadian study assessing the effect of childbirth reported no effect of pregnancy on the time taken to reach a confirmed EDSS 6, albeit they did not take into consideration treatment with disease-modifying agents [13]. More recently, a cross-sectional Spanish study investigated the effect of pregnancies on the risk of developing MS and long-term disability accrual using a multivariate approach based on 501 women with CIS. They found no protective effect of pregnancy on the conversion from CIS to definite MS, nor an effect of disability accrual on all other variables (clinical activity, MRI findings, DMDs). Pregnancies were protective in the univariate analysis, but the effect was lost in multivariate analyses when using a time-dependent statistical approach [14]. 

In 2020, the Israeli Multiple Sclerosis Pregnancy Study Group [29] performed a population-based retrospective cohort study that included 2281 females with relapsing–remitting MS, and reported that parity was associated with a decreased risk of progression to moderate disability (adj.HR, 0.68; 95% CI 0.54–0.85, *p* = 0.001), and hazard ratios for progression were comparable between women with one, two, and three or more births but the data showed a non-significant trend that second and third births decreased the probability of progressing to moderate disability. This study suggests that childbirth is a predictor of improved moderate disability-free survival, after adjustment for several risk factors, and indicates that giving birth after multiple sclerosis onset extends the time to moderate disability progression.

### 3.3. Menopause, Breastfeeding, HRT

MS worsens at menopause (particularly with the sudden decline after surgical menopause) due to the withdrawal of estrogen [12,52,53,54], and women treated with HRT note that their MS symptoms and quality of life improve [55]. 

Both cases described here reported a relatively long breastfeeding period. In MS, the role of breastfeeding related to disease activity remains controversial. Indeed, while a meta-analysis [56] and recent studies [31,57,58,59,60,61,62] showed that exclusive breastfeeding has a protective effect on clinical activity of MS, other findings favored a more neutral role and suggested a “reverse causality”, linking the choice to breastfeed with less active disease before and during pregnancy [24,63], given that relapse recurrence appeared to be predicted only by disease activity before and during pregnancy, while a lower risk of postpartum relapse was associated with the use of DMDs in the two years prior to conception [24,26].

However, to determine whether women who breastfeed their infants longer are at lower risk of developing MS, Langer Gould and colleagues recruited many women with newly diagnosed MS or its precursor and matched controls. They demonstrated that a cumulative duration of breastfeeding for longer than 15 months was associated with a reduced risk of MS/CIS (adjusted odds ratio [OR] 0.47, 95% CI 0.28–0.77; *p* = 0.003 compared to 0–4 months of breastfeeding), thus being at lower subsequent risk of developing MS [64].

Moreover, studies on prolactin (PRL), the peptide hormone which mainly increases after delivery, demonstrated that high PRL levels in MS patients may have a protective role [65]. Etemadifar M. and collaborators reported a lower disability level in patients with hyperprolactinemia compared with MS patients with normal hormone levels [66], while Jacob and colleagues showed an association between low levels of prolactin and risk of MS and suggested that low PRL may be used as a biomarker for MS progression [67]. In 2019, Krysko and colleagues confirmed by means of a large-scale systematic metanalysis that women who breastfed compared to women who did not were protected from post-partum relapses to the extent of a 43% decrease in relapse incidence if exclusive breastfeeding was considered, due to the PRL increase [68]. 

Pregnancy in women with MS is still considered to be a risk, although disease activity seems to decrease, and nowadays there is a significant rising trend in pregnancy rates among women with MS, unlike those that do not present the disease [69].

Our first patient reported the onset of symptoms soon after the discontinuation of HRT. The incidence of MS is 40% lower in oral contraceptive users compared with nonusers [70]. An interventional study supports the anti-inflammatory effects of high-dose estrogens on MRI findings and a significant improvement in cognitive performance [71,72]. Moreover, systemic HRT use is associated with a better physical quality of life in postmenopausal women with MS [55]. 

## 4. Conclusions

The two cases of multiparity reported here underline the potential protective role of pregnancy, breastfeeding, and HRT in delaying the onset of clinical symptoms related to MS.

The past decade has witnessed accumulating evidence of a beneficial effect of parity on the long-term disease course of MS, or at least no differences. No studies have shown that pregnancy could worsen the disease course. Determining the influence of pregnancy on the long-term outcomes of women with MS will have an enormous impact for women of childbearing age with MS, who are the majority of our patients. During the last few years, an increase in the percentage of pregnancies among MS women has been observed, in contrast with the well-known decrease in the global birth rate in Western countries. These findings suggest that clinicians are suggesting to more women with MS that they can experience motherhood [73].

Prospective studies on large cohorts of patients matched for age at the time of their MS diagnosis need to be performed in order to determine whether multiparity might have an impact on both clinical presentation and lesion burden at MRI. Further investigations on the potential impact of HRT in the perimenopausal management of MS women are needed. Only with this effort will we learn how to better advise and treat our patients and let them experience a fulfilling reproductive life and beyond, despite their MS. 

## Figures and Tables

**Figure 1 medicina-59-00619-f001:**
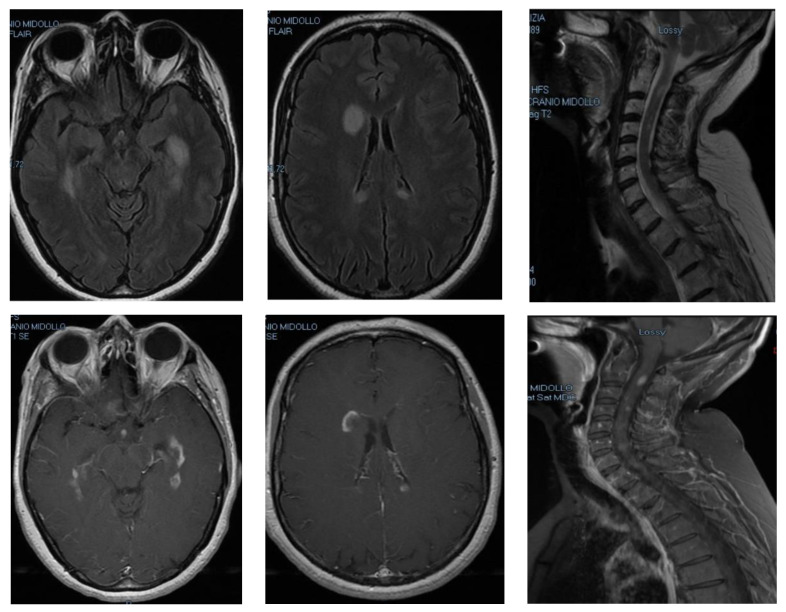
Case Report 1: T2-weighted sequences (top row) and T1-weighted post-gadolinium (bottom row) of brain and spinal cord MRI scan acquired at MS symptom onset.

**Figure 2 medicina-59-00619-f002:**
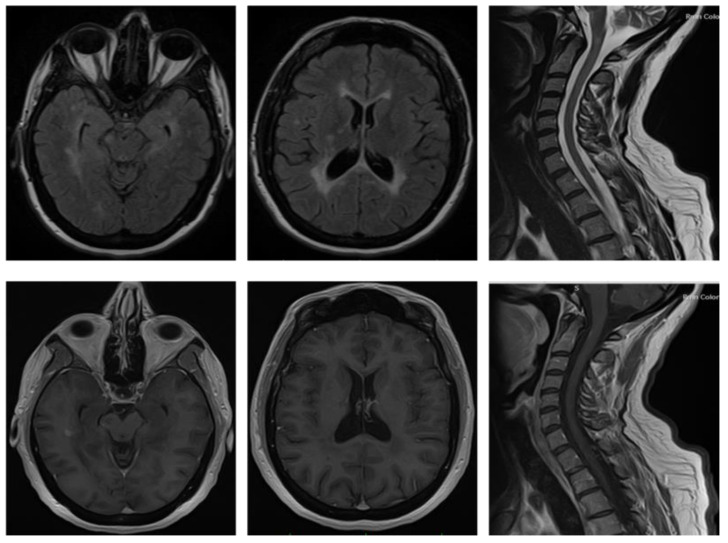
Case Report 2: T2-weighted sequences (top row) and T1-weighted post-gadolinium (bottom row) of brain and spinal cord MRI scan acquired at MS symptom onset.

**Figure 3 medicina-59-00619-f003:**
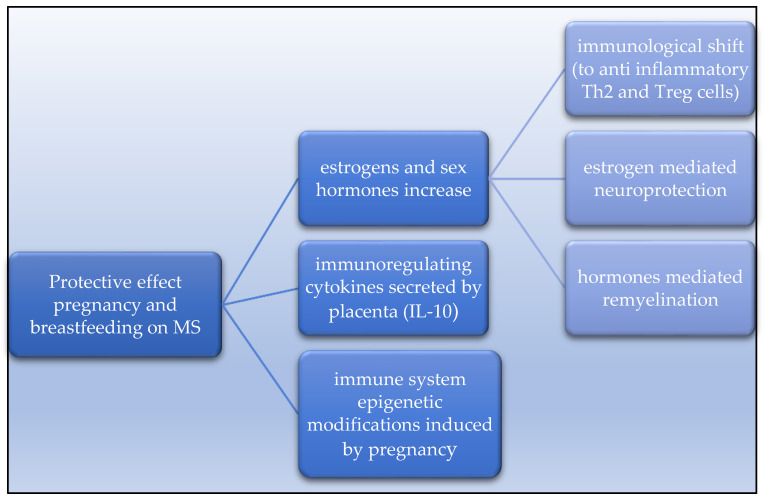
Mind map-like figure of the possible biological mechanisms that contribute to the potential protective role of pregnancy and breastfeeding in MS.

**Table 1 medicina-59-00619-t001:** Summarizing table of studies on the impact of pregnancy and breastfeeding in MS, comments and evidence provided.

Study	Evidence	Comments
Confavreux—PRIMS study 1998 [8]	ARR decreases during pregnancy and breastfeeding,	- First study ever to support a protective role of pregnancy - Conducted before the advent of DMDs
Confavreux—2000 [20]	In the first 3 months post-partum, disease activity correlates with the n. of relapses pre-pregnancy	- Post-partum period reflects disease activity pre-pregnancy
Vukusic—2004 [21]	Remarkable decrease in ARR during 3rd trimester, not different from the pre-pregnancy ARR	- 2 years post-partum follow up of the PRIMS study
Ebers—2008 [22]	ARR 12 months pre-pregnancy overall equals ARR in 9 months pregnancy + 3 post-partum	- Post-partum period reflects disease activity pre-pregnancy
Finkelsztein—2011 [23]	ARR decreases during pregnancy, MS does not correlate with obstetrical complications	- Large metanalysis - First to evaluate obstetrical outcomes
Portaccio—2011 [24]	Breastfeeding does not relate to post-partum relapses, while DMD administration is protective also in the post-partum period	- Post-partum period reflects disease activity pre-pregnancy and can be modified by DMDs
Teter—2013 [25]	Parous women take longer time to demonstrate disability progression than nulliparous women	- Cohort study on EDSS 6 milestone
Hughes—2014 [26]	ARR decreases during pregnancy, DMD administration is protective also in the post-partum period	- Protective role of pregnancy - Post-partum period reflects disease activity pre-pregnancy and can be modified by DMDs
Masera—2015 [12]	Parous women take longer time to demonstrate disability progression than nulliparous women	- Retrospective study on EDSS 4 and 6 milestones
Jokubaitis—2016 [11] and Houtchens—2018 [27]	ARR decreases during pregnancy more than it does with first-line DMDs	- Large systematic review - Performed with DMDs available at the time when women were advised not to take DMDs months before planning pregnancy
Dobson—2020 [28]	ARR decreases during pregnancy, and the early resumption of DMDs is protective also in the post-partum period	- Metanalysis on studies since PRIMS
Achiron—2020 [29]	Childbirth predicts moderate disability—free survival	- Retrospective study
Modrego—2021 [30]	ARR decreases during pregnancy compared to the year pre-pregnancy MS does not correlate with obstetrical complications	- Metanalysis including obstetrical outcomes
Hellwig—2021 [31]	DMD exposure before pregnancy decreases risk of relapse during pregnancy and post-partum	- Systematic review on DMDs - High-efficacy novel DMD withdrawal is not compensated by the protective effect of pregnancy on relapse

## Data Availability

Not applicable.

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
