# Peer review of "Potential Protective Role of Pregnancy and Breastfeeding in Delaying Onset Symptoms Related to Multiple Sclerosis"

_medicina, 2023, doi:10.3390/medicina59030619_

Round 1

Reviewer 1 Report

This manuscript is a case reports of two multiparous patients developing MS episodes. The study and connections drawn is interesting to the people in the field. There are only minor concerns as follow:

It will be good to summarise the past finding in a table to provide a clear and quick glance to the readers. A summative figure would also be helpful..

There are multiple mistakes detected throughout the manuscript. Examples are as follow but there are too many others to list.

Line 116: add space to "51years " 

Line 147: typo "tsmoking" 

Line 150: correct ".."

Author Response

Please see the attachment , thank You. 

Reviewer 2 Report

Abstract

We understand that cohort studies are needed, but we think it would be better to mention that "prospective" cohort studies are needed.

L66; "MS" is already abbreviated, so we do not believe spelling it out is necessary; please use the abbreviation in L72.

L93; "disease-modifying drugs (DMDs)" repeatedly spelled out and abbreviations used.

The following drugs are available for multiple sclerosis, but the authors mention only some of them. Is there a need to provide background safety information for pregnant and nursing women for other drugs?

Interferon beta (INF-beta)

fingolimod

Natalizumab

Glatiramer acetate

Dimethyl fumarate

Siponimod fumarate tablet

L132 Does "1 gr" mean "1 g"?

Do you have a description of when you diagnosed Case 1 with MS and the diagnostic criteria?

Does Case 1 represent a case that experienced pregnancy-birth-breastfeeding, meaning that the onset or worsening of the disease occurred after multiple deliveries?

Although the post-pregnancy transitions are well described, what were the pre-pregnancy circumstances?

Case 2 also needs to be considered in the same way.

Is there a need to report ARR in each case report? Objective symptoms need to be described.

Is there a need to provide a timeline of drug therapy in each case report?

In each case report, is there a need to examine the immunological examination of the patient?

What approach can we take in the future to scientifically elucidate the association between pregnancy and lactation and multiple sclerosis disease?

L241 "n a2-year post-partum 2" A space is needed after the "a".

L247 The way p-values are presented needs to be consistent throughout the paper.

L255 What is the difference between "average relapse rate during the year" and "ARR"?

L353 "Disease Modifying Therapies (DMDs)" If this is a first publication, this notation is acceptable, but I think it has already been published.

L361 "DMDs?"

In each section, could you add what interventions and medications are appropriate for patients in each setting (pre-pregnancy, menopause, etc.) rather than just summarizing previous research?

Author Response

please see the attachment, thank you

Reviewer 3 Report

In the Abstract, 

The following sentence needs to be rephrased "We performed a literature review to better characterize patients with similar characteristics, 27 since medical knowledge is constantly changing, and this topic remains controversial."

In the introduction,

Please, when using reference numbers, use a "space" between them and the word before. i.e., "Multiple sclerosis (MS) is a chronic demyelinating autoimmune neurodegenerative 45 disease of the central nervous system (CNS), usually occurring in young adults with a 46 female/male prevalence of approximately 3:1[1]."

Instead of repeatedly abbreviating MS, please add it to the abbreviations section and delete the entrances from the main text; it is redundant.i.e., "The knowledge regarding Multiple Sclerosis (MS)" and "Multiple sclerosis (MS) is a chronic demyelinating"

Please do not use the abbreviation section or abbreviate other terms in the main text using one or more methods.

Do not capitalize MS throughout the text: "At present, it is known that Multiple Sclerosis"

Also, a table is required in which the authors should present the findings from the literature in comparison with their findings as pro and con.

The manuscript would benefit from a mind-map-like figure in which the authors should explain the chain of events leading to the potential protective role of breastfeeding and pregnancy for MS women. 

By paying more attention to the English style of writing, it would be a very interesting article.

Author Response

please see the attachment, thank you

Round 2

Reviewer 2 Report

The points raised to the reviewers have been answered.